# Design of a Deployable Broadband Mesh Reflector Antenna for a SIGINT Satellite System Considering Surface Shape Deformation

**DOI:** 10.3390/s24020384

**Published:** 2024-01-08

**Authors:** Changhyeon Im, Wongu Seo, Seulgi Park, Kihun Kim, Sungkyun Park, Hosung Choo

**Affiliations:** 1Department of Electronic and Electrical Engineering, Hongik University, Seoul 04066, Republic of Korea; b615171@mail.hongik.ac.kr; 2Hanwha Systems Company Ltd., Seongnam 04541, Republic of Korea; swg0905@hanwha.com (W.S.); sg0212.park@hanwha.com (S.P.); kihun19.kim@hanwha.com (K.K.); sungkyun.park@hanwha.com (S.P.)

**Keywords:** satellite antenna, deployable reflector, mesh reflector, shape deformation

## Abstract

In this paper, we propose a deployable broadband mesh reflector antenna for use in signals intelligence (SIGINT) satellite systems, considering performance degradation due to shape deformation. To maximize gain by increasing the diameter of the reflector while reducing the weight of the antenna, the reflector of the antenna is designed using lightweight silver-coated Teflon mesh. The mesh reflectors are typically expanded by tension to maintain their parabolic structure; thus, shape deformation cannot be avoided. This shape deformation results in shape differences between the surface of the mesh reflector and the ideal parabolic reflector, thus resulting in the degradation of the performance of the mesh reflector antenna. To observe this degradation, we analyze antenna performance according to the number of arms, the number of joints, the feed distance, and the distance from the reflector center to each joint. The performance of the mesh reflector antenna is examined using an effective lossy conducting surface (ELCS) that has the same reflectivity as the silver-coated Teflon mesh to reduce simulation time and computing resources. The designed silver-coated Teflon mesh reflector and the double-ridged feed antenna are fabricated, and the bore-sight gain is measured using the three-antenna method. The measured bore-sight gain of the proposed antenna is 31.6 dBi at 10 GHz, and the measured and simulated results show an average difference of 3.28 dB from 2 GHz to 18 GHz. The proposed deployable mesh reflector antenna can be used in a variety of applications where small stowed volume is required for mobility, such as mobile high-gain antennas as well as satellite antenna systems. Through this study, we demonstrate that shape deformation of the mesh reflector surface significantly affects the performance of reflector antennas.

## 1. Introduction

In recent years, signals intelligence (SIGINT) satellites have been operated to analyze important information by collecting various signals generated from the earth’s surface [1,2,3,4]. By monitoring communication, electronic, and measurement signals, it is possible to perceive various conditions and prepare for potential threats more accurately [5,6,7]. To collect various signals, SIGINT satellites are required to be equipped with SIGINT antennas, which should have high-gain, high sidelobe level (SLL), and broadband characteristics. The high-gain characteristics are necessary to compensate for significant path losses resulting from long-distance and atmospheric losses [8,9,10,11]. The high SLL is an important requirement to maintain the high quality of the collected data and to avoid interference with signals from undesired regions. Broadband characteristics are required to cover various applications, such as communication intelligence (COMINT), electronic signals intelligence (ELINT), and foreign instrumentation signals intelligence (FISINT) [12,13,14,15]. To achieve these required antenna characteristics, parabolic reflector antennas are typically employed in many SIGINT systems. However, to achieve high-gain and high SLL characteristics, the diameter of the reflector should be increased, which precipitates a trade-off problem with limited payload bays during satellite rocket launches. Therefore, reducing the volume and weight while maintaining the necessary characteristics is the most important issue when stowing the reflector antenna on a satellite. To satisfy these requirements, many studies, such as compact reflector antennas with a minimized stowed volume [16,17], folded reflectors divided into multiple conductive panels [18,19,20,21], and mesh reflectors replacing the reflector surface with conductive meshes [22,23,24], have been investigated. These studies have generally focused on how well the curvature of the reflector can be mechanically deployed and how small the stowed volume is in the payload bay of the rocket. However, even if the deployable frame structure is well deployed mechanically, surface shape deformation of the mesh reflector cannot be avoided due to the tension required to maintain the parabolic structure [25,26,27]. There is not sufficient research concerning the degradation of the radiation characteristics associated with the surface shape’s deformation during the reflector deployment. It is also important to examine the antenna’s performance according to the frame structure of the mesh reflector, such as the number of arms, locations of the joints, and feed distance since shape deformation is caused by multiple flat pieces of mesh surfaces generated by the frame structure.

In this paper, a deployable broadband mesh reflector antenna is proposed for use in SIGINT satellite systems. To maximize gain by increasing the diameter of the reflector while reducing the weight of the antenna, the reflector of the antenna is designed using lightweight silver-coated Teflon mesh. The mesh reflectors are typically expanded by tension to maintain their parabolic structure; thus, shape deformation cannot be avoided compared to an ideal parabolic shape. The shape deformation of the deployable mesh reflector is closely related to elements of the frame structure, such as arms and joints. In this study, antenna performances in relation to shape deformation are closely investigated for the first time by varying the number of arms, the location of the joints, and the feed distance. To apply the proposed mesh reflector to the SIGINT system, which requires extremely broadband characteristics, a double-ridged horn antenna covering 2 GHz to 18 GHz is designed [28]. To obtain the high-gain characteristics of the mesh reflector, the optimal location of the wideband horn antenna according to the deformation of the mesh reflector is also examined. The performances of the mesh reflector and the feed horn are investigated using the CST Studio Suite full EM simulator (Dassault systems, Seattle, WA, USA) [29]. The total simulation time is increased significantly because the diameter of the mesh wire is very small (0.0007λ at 10 GHz) compared to the reflector diameter (33λ at 10 GHz). To reduce the simulation time and computing resources, the mesh surface of the reflector is replaced in the simulation by an effective lossy conducting surface (ELCS) that has the same reflectivity as the silver-coated Teflon mesh. To verify the performance of the mesh reflector antenna, the silver-coated Teflon mesh reflector and the double-ridged feed antenna are fabricated. Then, the bore-sight gain of the reflector antenna is measured using the three-antenna method. The measured bore-sight gain is 31.6 dBi at 10 GHz, and the measured and simulated results show an average difference of 3.28 dB from 2 GHz to 18 GHz. The proposed deployable mesh reflector antenna can be used in a variety of applications where small stowed volume is required for mobility, such as mobile high-gain antennas as well as satellite antenna systems. Through this study, we also demonstrate that shape deformation of the mesh reflector surface significantly affects the performance of reflector antennas.

## 2. Geometry of Mesh Reflector Antennas and Performance

Figure 1 illustrates the geometry of the deployable broadband mesh reflector antenna, which consists of the mesh reflector and the feed antenna, as shown in Figure 1a. The mesh reflector is designed with the frame structure and the conductive mesh. The frame structure includes multiple arms and joints to reduce the volume effectively when it is stowed in the payload bay of the rocket. The lightweight conductive mesh (silver-coated Teflon mesh) is then attached to the upper surface of the frame structure to operate as a reflector surface. The mesh reflector surface is typically expanded through the tension that occurs when the arms are extended, thus forming its parabolic shape. However, since the reflector is formed by multiple flat pieces generated by the frame structure, shape deformation cannot be avoided compared to an ideal parabolic shape. The detailed shape of the reflector is determined by the frame structure, such as multiple arms, joints, and the position of the feed antenna. The shape of the reflector has a significant impact on the radiation characteristics, such as gain, half-power beamwidth, and SLL. Therefore, to maximize the radiation performance, it is necessary to optimize the frame structure of the mesh reflector from the perspective of the reflector’s deformation. To optimize the surface shape of the reflector, we analyzed the antenna performance by varying the number of arms (*N_a_*), the number of joints (*N_j_*), and the feed distance (*d_f_*). To apply the proposed reflector antenna to the SIGINT system, which requires extreme broadband characteristics, a double-ridged horn antenna covering 2 GHz to 18 GHz is employed as the feed antenna. Figure 1b presents the stowed geometry of the mesh reflector, with a volume of 0.016 m^3^, demonstrating an 89.5% reduction in volume compared to the fully deployed shape (0.149 m^3^). This reduction in volume allows it to be easily stowed into the payload bay of a rocket while maintaining high antenna gain with a fully deployed configuration. Figure 1c illustrates the side view of the proposed antenna. The curvature of this reflector is determined by the following parabolic Equation (1): (1)$$F=\frac{{\left(D_{1}/2\right)}^{2}}{4\times f}$$
which is conventionally used for designing reflector antennas. In this equation, *D*_1_ represents the diameter of the reflector when it is fully deployed, and *f* is the ideal focal point of the parabolic equation. The focal point *f* is a single spot where all incoming radio waves are concentrated based on the parabolic equation, and *f*/*D*_1_ of 0.5 is used for the proposed mesh reflector. The location of the joint on the *x*-axis is represented as *d_jn_*, which is the distance from the reflector center to the *n^th^* joint. The detailed parameters are then optimized to obtain high-gain, high SLL, and broadband characteristics, and they are listed in Table 1.

Figure 2 presents photographs of the fabricated reflector antenna and the measurement setup. Figure 2a illustrates the frame structure of the mesh reflector, which is fabricated in three parts (central hub, arm_1_, arm_2_) through plastic 3D-printing and then assembled for a complete structure. On the upper surface of the frame, the silver-coated Teflon mesh is attached using an adhesive, as shown in Figure 2b. Figure 2c presents the fabricated double-ridged horn antenna employed as the feed antenna. This antenna is fabricated using direct metal laser sintering (DMLS) 3D printing technology and operates from 2 GHz to 18 GHz. Figure 2d shows the measurement setup to obtain the bore-sight gain using the three-antenna method [30]; this method is frequently used to obtain an antenna gain (*G*), as expressed in Equation (2).
(2)$$G=(S_{21}^{AB}+S_{21}^{AC}-S_{21}^{BC}-P_{L})/2\;P_{L}=10\log_{10}({\left(\lambda /(4\pi d)\right)}^{2})$$

In the equation, $S_{21}^{AB}$, $S_{21}^{AC}$, and $S_{21}^{BC}$ represent transmission coefficients between antennas A and B, A and C, and B and C, respectively. The path loss (*P_L_*) is calculated using the wavelength (*λ*) and the distance (*d*) between the antennas. Figure 3 illustrates the bore-sight gains of the proposed antenna obtained using Equation (2), wherein the ‘×’ marks and the dashed line indicate the measured and simulated results. The bore-sight gains at the center frequency of 10 GHz are 31.6 dBi by measurement and 34.5 dBi by simulation. The measured and simulated results show an average difference of 3.28 dB from 2 GHz to 18 GHz. The bore-sight gain of the ideal reflector antenna can be calculated by Equation (3) [30,31]. It is observed that the bore-sight gain is reduced compared to the ideal case due to the shape deformation and mesh reflectivity.
(3)$$G_{ideal}(dB)=10\log_{10}6{\left(\frac{D_{1}}{\lambda }\right)}^{2}$$

## 3. Analysis

Figure 4a presents a photograph of the silver-coated Teflon mesh used for the reflector surface, and Figure 4b is a zoomed-in image of the mesh, showing the detailed lattice structure. This mesh has an opening per inch (OPI) of 20, meaning that there are 20 openings per inch, and it exhibits a reflectivity exceeding 97%. The minimum requirement for the reflectivity of the mesh reflector is typically 97%. The performance of the mesh reflector antenna is then examined using the CST Studio Suite full EM simulator [26]; however, the total simulation time is increased significantly since the entire structure is excessively large compared to the wavelength. To reduce the simulation time and computing resources, the mesh surfaces of the reflector are replaced in the simulation with an effective lossy conducting surface (ELCS) that has the same reflectivity as the silver-coated Teflon mesh.

In Figure 5a, the ‘×’ marks indicate the reflectivity of the ELCS, and the dashed line represents the simulated result for the actual Teflon mesh shape. The dotted line indicates the minimum reflectivity requirement (typically 97%) of the mesh surface to minimize degradation of antenna gain. When the reflectivity is set to 97%, the gain reduction is less than 0.1 dB compared to the reflector with a PEC surface. To analyze the reflectivity, we model the grid structure of the silver-coated Teflon mesh and confirm that the reflectivity is over 97% in the frequency range from 2 GHz to 18 GHz. We then determine the conductivity of the ELCS, which matches the reflectivity of the Teflon mesh at each frequency; for example, to achieve the same reflectivity with the actual Teflon mesh shape at 6 GHz and 10 GHz, the conductivities of the ELCS are set to 0.012 × 10^7^ S/m and 0.0044 × 10^7^ S/m, as shown in Figure 5b. The conductivity of the ELCS for each frequency is derived by comparing the reflectivity of the ELCS with the reflectivity of the actual Teflon mesh shape.

The mesh reflectors are typically expanded by tension to maintain their parabolic structure; thus, shape deformation cannot be avoided. The shape deformation of the deployable mesh reflector is closely related to elements of the frame structure, such as arms and joints. Therefore, a case study is conducted by varying the number of arms in the frame structure (*N_a_*), the number of joints (*N_j_*), the feed distance (*d_f_*), and the distance of the joints (*d_j_*_1_, *d_j_*_2_) to obtain high-gain characteristics.

Figure 6 illustrates the bore-sight gain in accordance with variations of *N_a_* when *f/D*_1_ of the antenna is 0.5 at 10 GHz. As shown in the result, when *N_a_* is less than 14, the bore-sight gain increases rapidly as *N_a_* is raised; however, when *N_a_* exceeds 14, the rate of increase gradually declines and saturates.

Figure 7 represents the bore-sight gain and SLL in accordance with variations of *N_a_* and *N_j_*, while maintaining *d_f_/D*_1_ as 0.5 at 10 GHz. In Figure 7a, the blue and red lines indicate bore-sight gain and SLL, respectively. As mentioned above, when *N_a_* is less than 14, the bore-sight gain increases as *N_a_* increases, while there is almost no difference in the bore-sight gain when *N_a_* exceeds 14. In contrast, the SLL is more sensitive to changes in *N_a_*, exhibiting a larger fluctuation according to *N_a_* than the bore-sight gain; it remains at a constant value when *N_a_* exceeds 24. Considering these performance changes, the minimum *N_a_* value that can simultaneously satisfy both high-gain and high SLL characteristics is 24, wherein the bore-sight gain is 35.6 dBi and the SLL is 27.6 dB. Figure 7b indicates the bore-sight gain and the SLL in accordance with *N_j_*. When *N_j_* exceeds 7, the bore-sight gain and the SLL do not increase significantly and have constant values of 36.5 dBi and 35.7 dB, respectively. Based on these results, the optimal *N_j_* value for deriving the high-gain and high SLL characteristics of the mesh reflector antenna considering shape deformation is 7. However, when the optimal *N_a_* of 24 and the *N_j_* of 7 are applied to the mesh reflector, it has a total of 168 joint structures, which increases the mechanical complexity and the possibility of mechanical errors. Therefore, in the proposed antenna, to reduce mechanical errors by minimizing the number of joints, optimization is performed to achieve high-gain characteristics with only two joints.

Figure 8 illustrates the bore-sight gain in accordance with variations in the distances of *d_j_*_1_ and *d_j_*_2_. When *d_j_*_1_ is 0.16 m and *d_j_*_2_ is 0.31 m, the bore-sight gain is 36 dBi, which is 0.3 dB higher than when located at a point divided by the same distance (*d_j_*_2_ = 2 × *d_j_*_1_, *d_j_*_1_ = 0.165 m, and *d_j_*_2_ = 0.33 m). Through these analyses, it is confirmed that the most critical variable in changing the antenna performance is the number of arms (*N_a_*) when designing a deployable mesh reflector antenna considering the shape deformation. In addition, it is confirmed that even when the number of arms is fixed, additional performance improvements are possible by changing the number of joints (*N_j_*) and the distance of each joint (*d_jn_*).

In Table 2, the proposed antenna is compared with previous studies [16,17,18,19,22,23]. In [16,17], since a single solid conducting surface is used as a reflector, the antenna cannot be folded, making it difficult to mount on a rocket. In [18,19], deployability can be achieved by using multiple conductive panels for the reflector surface, but these reflectors are heavy and have limitations in reducing the volume. To solve this problem, research on reflector antennas using mesh that is light and easy to change shape is being actively conducted [22,23]. However, it is only focused on mechanical analysis, and the radiation characteristics have not been verified through measurement.

In this work, mesh is used as the reflector surface, and the radiation characteristics are investigated for the first time by considering shape deformation. The proposed deployable mesh reflector antenna can be used in a variety of applications that require small stowed volume, such as mobile high-gain antennas and satellite antenna systems.

## 4. Conclusions

We proposed the deployable broadband mesh reflector antenna for use in SIGINT satellite systems, considering performance degradation due to shape deformation. To maximize gain by increasing the diameter of the reflector while reducing the weight of the antenna, the reflector of the antenna was designed using lightweight silver-coated Teflon mesh. The mesh reflectors were typically expanded by tension to maintain their parabolic structure; thus, shape deformation could not be avoided. This shape deformation resulted in a shape difference between the surface of the mesh reflector and the ideal parabolic reflector, resulting in the degradation of the performance of the mesh reflector antenna. To observe this degradation, we analyzed antenna performances according to the number of arms, the number of joints, the feed distance, and the distance from the reflector center to each joint. The performance of mesh reflector antennas was examined using ELCS, which has the same reflectivity as the silver-coated Teflon mesh, to reduce simulation time and computing resources. The designed silver-coated Teflon mesh reflector and the double-ridged feed antenna were fabricated, and the bore-sight gain of the reflector antenna was measured using the three-antenna method. The measured bore-sight gain of the proposed antenna was 31.6 dBi at 10 GHz, and the measured and simulated results showed an average difference of 3.28 dB from 2 GHz to 18 GHz. In this study, mesh was used as a reflector surface, and its radiation characteristics were analyzed considering shape deformation. The antenna’s performance in relation to shape deformation was closely investigated. In order to verify the performance of the proposed antenna, the reflector antenna, including the feeder, was fabricated and measured. The proposed deployable mesh reflector antenna can be used in a variety of applications where small stowed volume is required for mobility, such as mobile high-gain antennas as well as satellite antenna systems. Through this study, we demonstrated that shape deformation of the mesh reflector surface significantly affects the performance of reflector antennas.

## Figures and Tables

**Figure 1 sensors-24-00384-f001:**
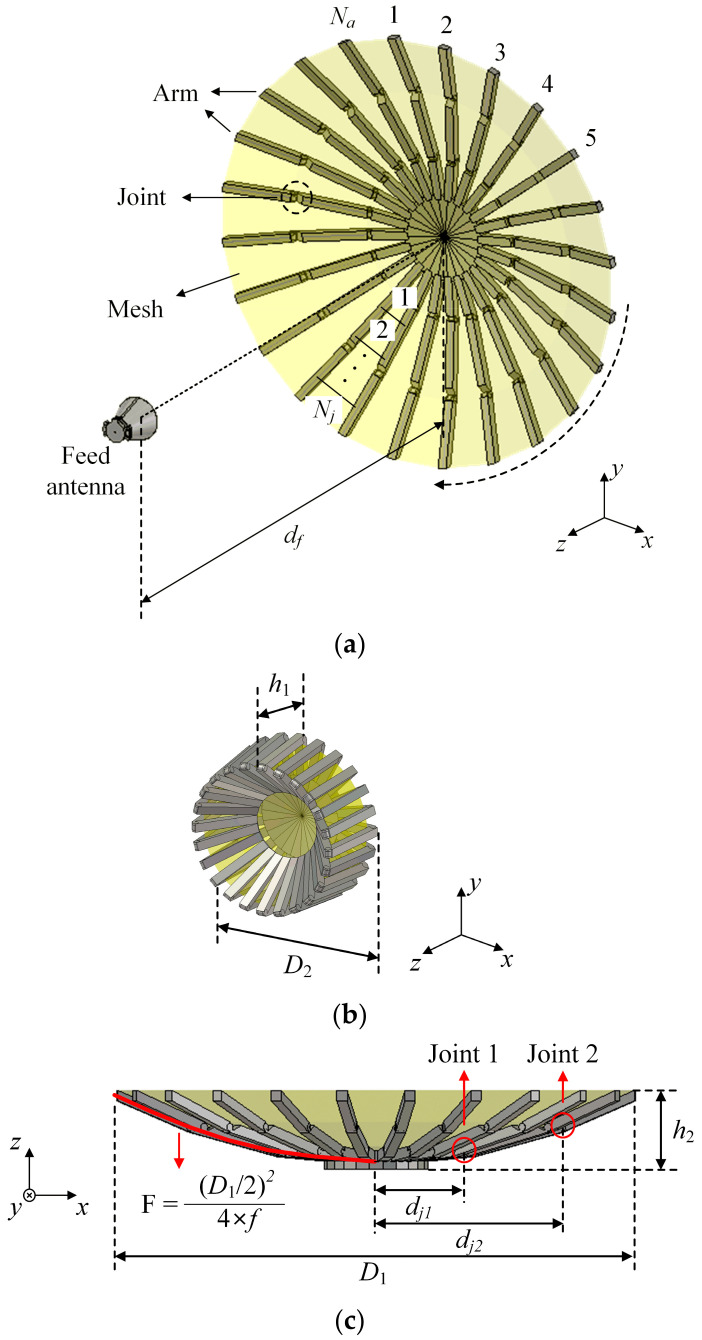
Geometry of the mesh reflector antenna: (**a**) isometric view (deployed shape); (**b**) isometric view (stowed shape); (**c**) side view (fully deployed shape).

**Figure 2 sensors-24-00384-f002:**
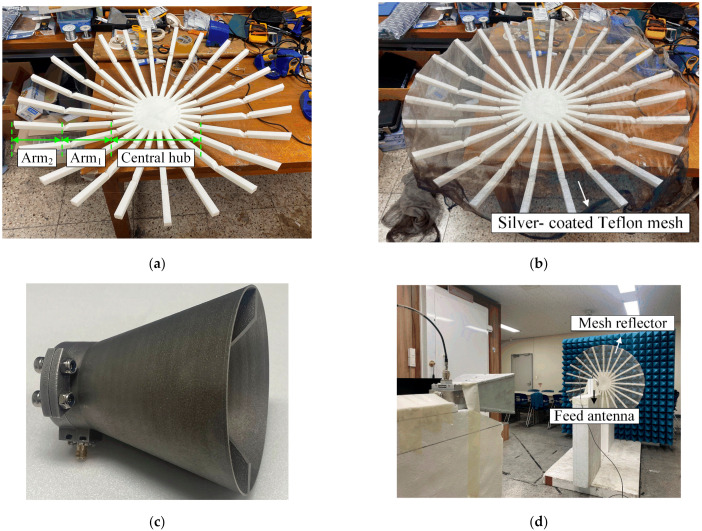
Photographs of the fabricated reflector antenna and measurement setup: (**a**) reflector frame without mesh; (**b**) reflector frame with mesh; (**c**) double-ridged horn antenna; (**d**) measurement setup.

**Figure 3 sensors-24-00384-f003:**
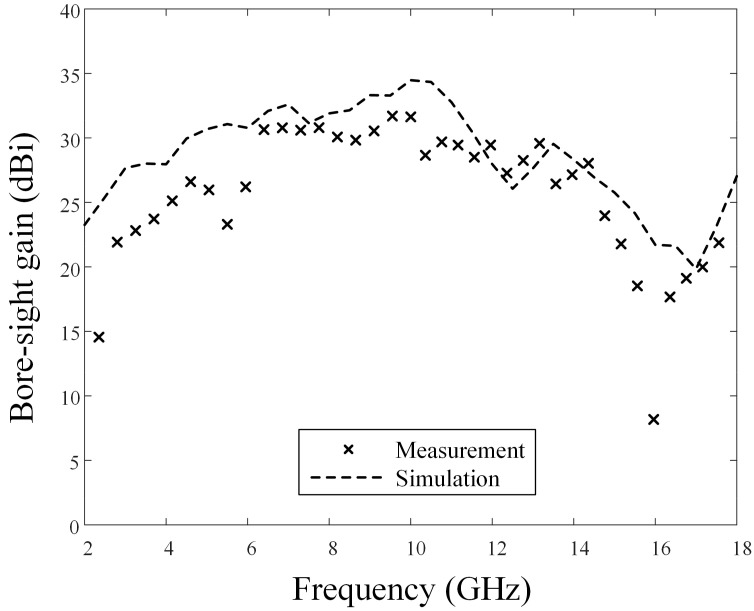
Measured and simulated bore-sight gain of the proposed antenna.

**Figure 4 sensors-24-00384-f004:**
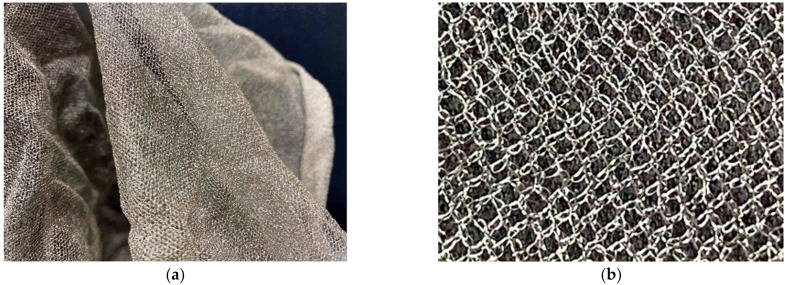
Photographs of the silver-coated Teflon mesh: (**a**) silver-coated Teflon mesh; (**b**) silver-coated Teflon mesh (zoomed-in image).

**Figure 5 sensors-24-00384-f005:**
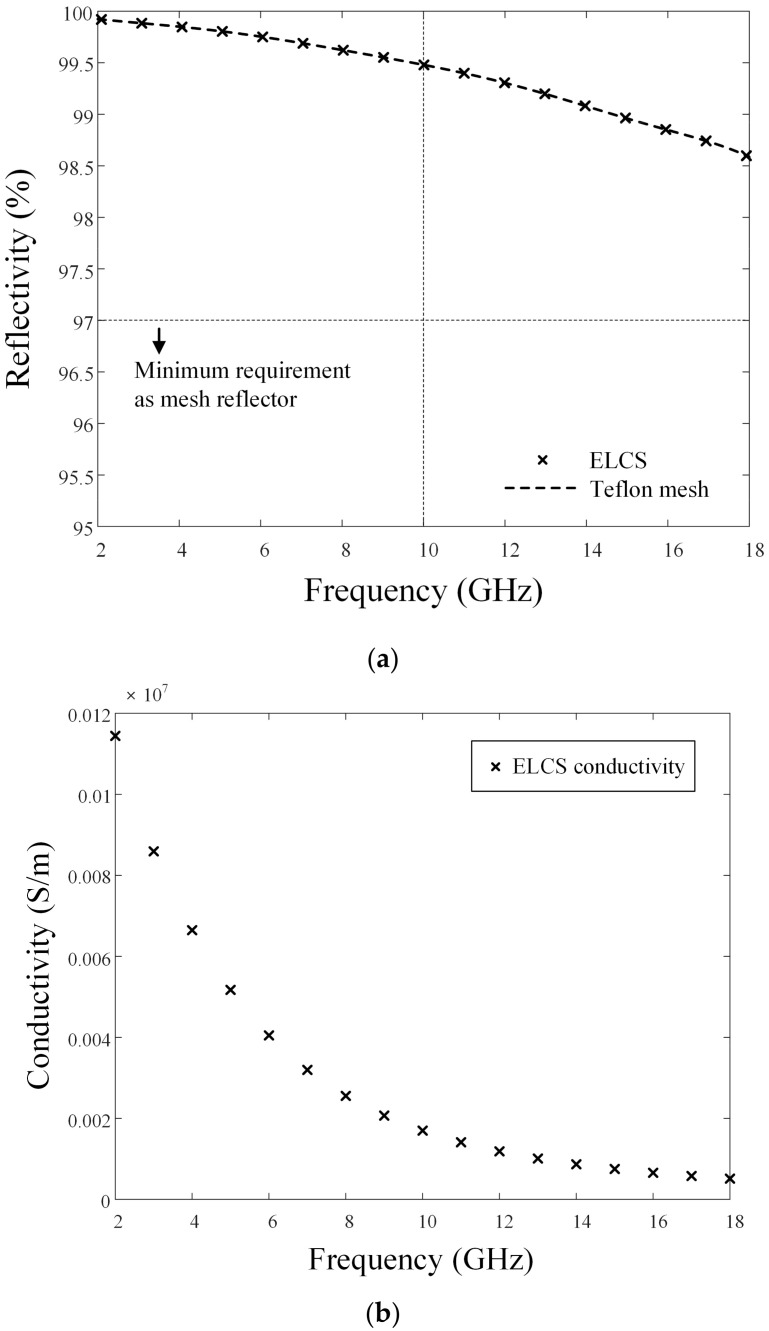
Reflectivity and conductivity of ELCS: (**a**) reflectivity; (**b**) conductivity.

**Figure 6 sensors-24-00384-f006:**
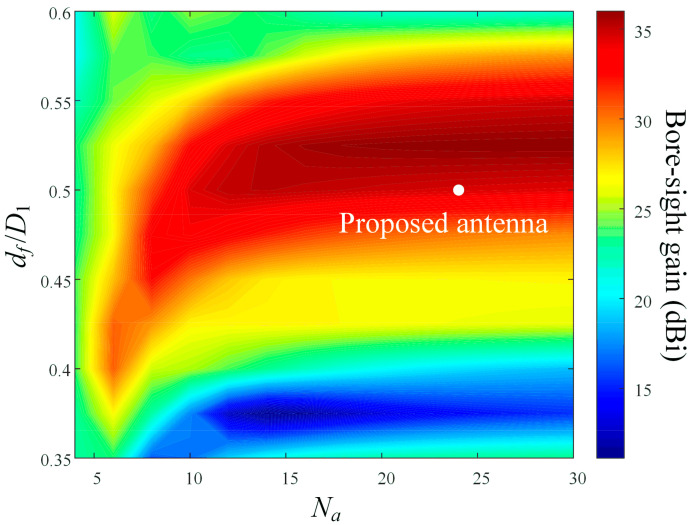
Bore-sight gain in accordance with variations of *N_a_* and *d_f_*/*D*_1_ at 10 GHz.

**Figure 7 sensors-24-00384-f007:**
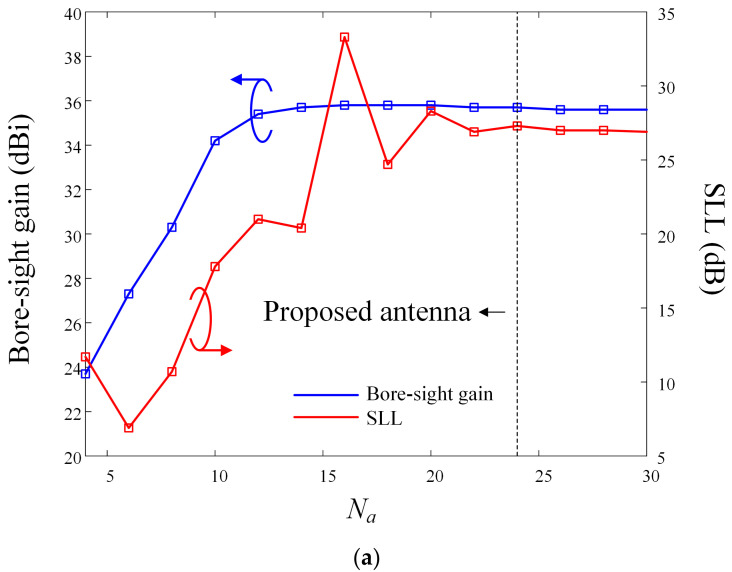
Bore-sight gain and SLL in accordance with variations of *N_a_* and *N_j_* (*d_f_/D*_1_ = 0.5) at 10 GHz: (**a**) *N_a_*; (**b**) *N_j_*.

**Figure 8 sensors-24-00384-f008:**
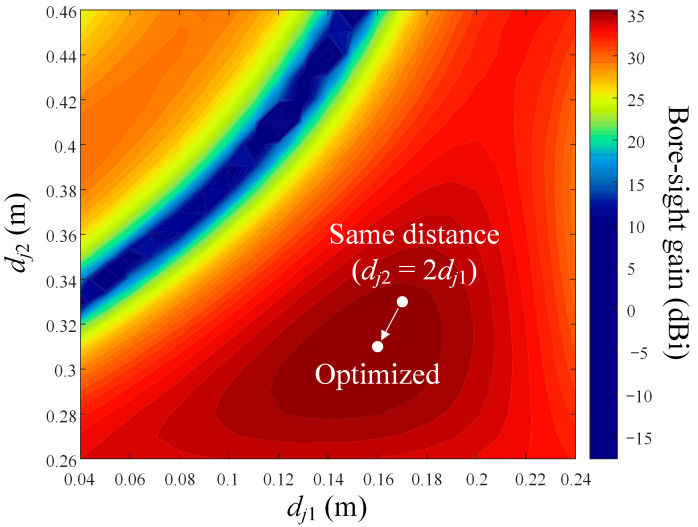
Bore-sight gain in accordance with variations of *d_j_*_1_ and *d_j_*_2_ at 10 GHz.

**Table 1 sensors-24-00384-t001:** Optimized design parameters of the proposed antenna.

Parameters	Values	Parameters	Values
*N_a_*	24	*h* _1_	0.19 m
*N_j_*	2	*h* _2_	0.125 m
*d_f_*	0.5 m	*d_j_* _1_	0.16 m
*D* _1_	1 m	*d_j_* _2_	0.31 m
*D* _2_	0.4 m	*f*	0.5 m

**Table 2 sensors-24-00384-t002:** Comparisons between the proposed antenna and some previous studies.

Research	Reflector Surface Material	Deployability	Analysis of Radiation Characteristics	Performance Verification
[16]	Solid conducting surface	×	O	Fabrication and measurement
[17]	Solid conducting surface	×	O	Fabrication and measurement
[18]	Solid conducting surface	O	×	Fabrication and measurement
[19]	Solid conducting surface	O	×	Fabrication and measurement
[22]	Mesh	O	×	Simulation only
[23]	Mesh	O	×	Simulation only
This work	Mesh	O	O	Fabrication and measurement

## Data Availability

Data are contained within the article.

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
