# Peer review of "Design of a Deployable Broadband Mesh Reflector Antenna for a SIGINT Satellite System Considering Surface Shape Deformation"

_sensors, 2024, doi:10.3390/s24020384_

Round 1
Reviewer 1 Report
Comments and Suggestions for Authors
In this paper, the authors propose a novel design of a deployable broadband mesh reflector antenna for SIGINT satellite systems, focusing on maximizing antenna gain while addressing the challenges posed by shape deformation due to the minimized number of joints in its structure.
In particular, the paper mentions using optimization to reduce joint count and enhance gain in the antenna design (as stated in 'Therefore, in the proposed antenna, to reduce the mechanical errors by minimizing the number of joints, optimization is performed to achieve high-gain characteristics with only two joints, as shown in Figure 8.'). However, it lacks a detailed description of the optimization algorithm used. Providing specifics about the algorithm, such as its type and approach, would enhance understanding of the design process. Thank you
Author Response
We would like to thank the reviewers for their constructive comments and for taking the time to consider our paper. We have tried our best to revise the paper in accordance with the reviewers’ comments. Our responses are detailed below. Modified parts in the paper are shown in red so that reviewers can easily see the changes. Please see the attachment.

Reviewer 2 Report
Comments and Suggestions for Authors
This paper investigated a deployable broadband mesh reflector antenna for signal intelligence satellite systems and investigated the electrical performance of the antenna with different numbers of arms through experimental measurement. My comments are listed as follows:
1. The proposed mesh reflector antenna is not an innovative structure since some umbrella antennas have been in orbit. The authors should make a literature review of those existing umbrella antennas similar to this paper.
2. This paper is actually to study the surface accuracy or surface error of mesh reflector antennas. Some related works focusing on the surface accuracy of mesh reflector antennas may also be added in the INTRODUCTION. The following references will hopefully help you.
https://doi.org/10.1016/j.actaastro.2014.07.029
Comments on the Quality of English LanguageNone
Author Response

(The authors gave the same response as above.)

Reviewer 3 Report
Comments and Suggestions for Authors
In this communication, a deployable broadband mesh reflector antenna for use in signals intelligence (SIGINT) satellite systems considering performance degradation due to shape deformation is proposed, which maximizes gain by increasing the diameter of the reflector while reducing the weight of the antenna.
However, to be accepted for publication, the following work should be done as suggested:
1. One of the main concern is the motivation and contribution of this work, whether this work is motivated by the similar work of previous reserchers, or it is entirely a new idea.
2. There lack a comparison of this work with the similar work, whether on performance or experiments results
3. The last paragraph of the Introduction part is terribly designed, which requires carefully modification.
4. The theoretical analysis of the paper should be enhanced, especially the equation and results analysis part.
5. Please clarify the future concrete utilization of this communication.
Comments on the Quality of English LanguagePlease check the English spelling as well as grammar, especially in the abstract, introduction, and conclusion parts.
Author Response

(The authors gave the same response as above.)

Reviewer 4 Report
Comments and Suggestions for Authors
This manuscript proposed a deployable broadband mesh reflector antenna and analyzed the performance degradation due to shape deformation. The demonstration of the concept is comprehensive, including analytical analysis and experimental results. Also, the paper is well organized, with step-by-step demonstrations. Therefore, I have only several comments for this manuscript.
(1) The advantages of the proposed deployable broadband mesh reflector antenna over other designs are not described clearly. Please add some comparisons or discussion in the background.
(2) Please revise the manuscript carefully, particularly on the Abstract and Conclusions.
(2) The format of the upper and lower corners in the formula is confusing, such as on page 3. Please retype all formulas and make the PDF version much better.
(3) The reflector's diameter, when fully deployed, is denoted as D on page 3. In Figure 1, this diameter is referred to as D2, also used in the analysis on page 7. It is recommended to use a consistent variable to represent this diameter.
(4) Using vector graphics or high-resolution images is recommended for displaying coordinate diagrams, such as Figure 3, Figure 5, Figure 6, Figure 7, and Figure 8.
(5) A comparison table with similar works should show the proposed design's advantages and disadvantages.
Comments on the Quality of English Language
The English of the manuscript is OK for me.
Author Response

(The authors gave the same response as above.)

Round 2
Reviewer 2 Report
Comments and Suggestions for Authors
I have no more questions.
Author Response
We would like to thank the reviewers for their constructive comments and for taking the time to consider our paper.
Reviewer 3 Report
Comments and Suggestions for Authors
The manuscript has been modified and improved, however, there are still some suggestions for your reference.
1. Some of the references are out of date, for example, [7]Matthew, M.A.; Cees, W. Introduction on the importance of signals intelligence in the cold war. Intell. Natl. Secur. 2001, 16, 1- 300 26.
2. The future application aspect should be addressed, especially in the abstract, Introduction and conclusion part.
3. The title of the paper is somewhat verbose, especially the latter part, "considering... due to ...".
4. In fig.5(a), there is a minimum requirement as mesh reflector, please explain the priciple.
5. The paragraph starting from line 205 should be separated, as it contains too much content and lacks some main idea sentences. Furthermore, whether the figures and tables of this paragraph have a relationship? If so, please discuss it and give some conjunctions or transitions.
6. The discussion of this work with previous literature from line 239 is not enough.
7. Please highlight the main contribution of this work, as this is a "communication". Furthermore, why is this work important to the LEO satellite? Please add more discussion at the end of the experiment and conclusion part.
Comments on the Quality of English Language
Moderate editing of English language is required, especially the Introduction and conclusion part.
Author Response

(The authors gave the same response as above.)
